# The Unfolded Protein Response and Its Implications for Novel Therapeutic Strategies in Inflammatory Bowel Disease

**DOI:** 10.3390/biomedicines11072066

**Published:** 2023-07-23

**Authors:** Noel Verjan Garcia, Kyung U. Hong, Nobuyuki Matoba

**Affiliations:** 1UofL Health—Brown Cancer Center, University of Louisville School of Medicine, Louisville, KY 40202, USA; noel.verjangarcia@louisville.edu (N.V.G.); kyung.hong@louisville.edu (K.U.H.); 2Department of Pharmacology and Toxicology, University of Louisville School of Medicine, Louisville, KY 40202, USA; 3Center for Predictive Medicine, University of Louisville School of Medicine, Louisville, KY 40202, USA

**Keywords:** endoplasmic reticulum stress, unfolded protein response, EPICERTIN, intestinal homeostasis

## Abstract

The endoplasmic reticulum (ER) is a multifunctional organelle playing a vital role in maintaining cell homeostasis, and disruptions to its functions can have detrimental effects on cells. Dysregulated ER stress and the unfolded protein response (UPR) have been linked to various human diseases. For example, ER stress and the activation of the UPR signaling pathways in intestinal epithelial cells can either exacerbate or alleviate the severity of inflammatory bowel disease (IBD), contingent on the degree and conditions of activation. Our recent studies have shown that EPICERTIN, a recombinant variant of the cholera toxin B subunit containing an ER retention motif, can induce a protective UPR in colon epithelial cells, subsequently promoting epithelial restitution and mucosal healing in IBD models. These findings support the idea that compounds modulating UPR may be promising pharmaceutical candidates for the treatment of the disease. In this review, we summarize our current understanding of the ER stress and UPR in IBD, focusing on their roles in maintaining cell homeostasis, dysregulation, and disease pathogenesis. Additionally, we discuss therapeutic strategies that promote the cytoprotection of colon epithelial cells and reduce inflammation via pharmacological manipulation of the UPR.

## 1. Endoplasmic Reticulum Stress (ER Stress)

The endoplasmic reticulum (ER) is a multifunctional organelle consisting of the nuclear envelope and the rough and smooth ER [1]. It performs a variety of cellular processes, including sterol and lipid biosynthesis, Ca^2+^ storage, and folding of newly synthetized proteins. Disruption of these processes negatively impacts the ER homeostasis, leading to the accumulation of misfolded/unfolded proteins (proteotoxicity) in the ER lumen, a condition called ER stress [2,3,4]. Upon sensing ER stress, the cell activates signaling pathways known as the unfolded protein response (UPR) to restore ER homeostasis [5].

The ER plays a key role in lipid membrane biogenesis. This involves the synthesis of enzymes required for the production of neutral lipids, such as triglycerides, cholesterol esters, and sphingolipids, that are incorporated into lipid membranes [6]. ER homeostasis is disturbed by factors such as the accumulation of exogenous saturated fatty acids, deficiency of desaturase enzymes, or diet-induced lipid depletion. These disruptions can cause lipotoxicity, followed by misfolded protein accumulation in the ER, and consequently, ER stress [7,8]. In addition, changes in the lipid composition of the ER membrane (saturated fatty acyl chains) increase the stiffness of the ER membrane, causing lipid bilayer stress [9]. This can directly activate the UPR without the involvement of unfolded proteins [10].

The ER serves as a reservoir for Ca^2+^ and regulates Ca^2+^ signaling through inositol 1,4,5-triphosphate (IP_3_) receptors (IP_3_R) and ryanodine receptors. These receptors are located in regions enriched with signaling proteins that are in contact with mitochondria, called mitochondria-associated ER-membrane (MAM) [3,4,11,12]. A constant level of Ca^2+^ in the ER lumen is essential for keeping Ca^2+^ receptors in a sensitive state [4] and supporting protein folding through Ca^2+^-dependent chaperones, such as calnexin (CANX), calreticulin (CALR), and heat shock protein family A member 5 (HSPA5), also known as immunoglobulin heavy chain-binding protein (BiP/GRP-78) [3]. Hence, a decrease in ER luminal Ca^2+^ may result in the accumulation of misfolded proteins, inducing ER stress.

Folding and assembly of newly synthesized proteins take place in the ER. This process ensures that only properly folded polypeptides proceed through the secretory pathway to their final cellular destinations [13]. In contrast, incompletely folded polypeptides are transported back to the cytosol for subsequent ubiquitylation and degradation by the 26S proteasome [14,15]. In conditions where there is an increased demand for protein synthesis, whether due to physiological needs or pathological conditions, the ER may accumulate misfolded proteins within its lumen, leading to ER stress [16].

Interruption of protein transport between organelles can also cause the accumulation of misfolded proteins in the ER, thus inducing ER stress. An example of this can be observed in brefeldin A-mediated inhibition of guanine nucleotide-exchange factors and vesicle trafficking [17,18]. Vesicle trafficking between the ER and Golgi is primarily regulated by seven transmembrane KDEL receptors (KDELRs 1–3) [19]. ER chaperones possesses a carboxyl-terminal Lys-Asp-Glu-Leu (KDEL) retrieval signal that binds to KDELRs in intermediate compartments and cis-Golgi, enabling their return to the ER via coat protein complex I (COPI)-coated vesicles [20,21]. Upon binding to chaperones and other KDEL-containing proteins in the Golgi, KDELRs become activated. This leads to the activation of heterotrimeric G proteins such as Gαq, which targets phospholipase C (PLC) for the generation of IP_3_ and diacylglycerol, and Gαs, which stimulates adenylate cyclase [22]. This process activates protein kinase A and Src family tyrosine kinases that mediate the phosphorylation of transport proteins to maintain homeostasis of the membrane transport apparatus [23,24,25]. Additionally, KDELR signaling activates cAMP response element binding protein 1, a transcription factor that upregulates genes involved in vesicle transport [26]. Through these mechanisms, KDELR activation helps maintain cell homeostasis by integrating transduction cascades with membrane trafficking, cytoskeleton reorganization, invadopodia (actin-based structures that facilitate extracellular matrix degradation and cancer cell invasion) formation, and remodeling of the extracellular matrix [25,27]. Conversely, impaired KDELR-mediated recycling of chaperones can cause their secretion into the extracellular medium and subsequent shortage in the ER [28]. This imbalance can result in increased accumulation of misfolded proteins in the ER, aggravating the ER stress and the UPR, which can have detrimental effects. For example, cells stably expressing a non-functional transport mutant KDELR (D193N) restricted reverse transport of COPI from the Golgi to the ER and became sensitive to ER stress. Transgenic mice expressing this mutant KDELR developed myocardial cell death and cardiac hypertrophy, and ultimately died due to heart failure [29]. These findings illustrate the consequences of disturbing the recycling of proteins between the ER and the Golgi complex, leading to the accumulation of misfolded protein and ER stress. Similarly, gene deletion or homozygous mutation of chaperone genes (HSPA5, CALR) affected heart physiology and were lethal [30,31,32].

In summary, disturbance in ER homeostasis, such as reduced calcium levels, increased misfolded proteins, or stiffness in the ER membrane, cause ER stress. If left unmitigated, this stress can have detrimental effects on cell physiology, leading to various pathological conditions and diseases.

## 2. The Unfolded Protein Response (UPR)

The UPR serves as a safeguard mechanism designed to adapt to physiological or pathological demands in protein synthesis [33] via the generation of effector molecules that control gene transcription, mRNA translation, and degradation of misfolded proteins [34]. The UPR is initiated by three highly conserved signal transduction machineries held within the ER membrane upon sensing ER stress. They include two type I transmembrane kinases—the ER transmembrane inositol-requiring enzyme 1α and 1β (IRE1α and IRE1β) [35] and the eukaryotic translation initiation factor 2 alpha kinase 3 (EIF2AK3/PERK) [36]—as well as one unprocessed transcription factor: activating transcription factor 6 (ATF6) [37].

### 2.1. IRE1α/β

IRE1α and IRE1β are encoded by *ERN1* and *ERN2* genes, respectively. IRE1α is ubiquitously expressed, whereas IRE1β is predominantly found in the mucosal epithelium [38]. IRE1 has an endoribonuclease (RNase) domain and a serine/threonine kinase domain, both of which participate in the UPR. The RNase activity of IRE1 mediates the splicing of a 26-basepair intron from the mRNA encoding X-box-binding protein 1 (XBP1), generating the spliced form (XBP1s). XBP1s activates the transcription of genes required for energy expenditure, metabolism, ER function (e.g., chaperones and KDELRs) [28], cell survival, and differentiation in a cell type specific manner [39]. Importantly, XBP1s mitigates ER stress and promotes ER function by controlling the transcription of ER factors required for protein folding (e.g., protein disulfide isomerase, PDI), secretion, and factors involved in degradation of misfolded proteins, which is known as the ER-associated degradation (ERAD) machinery. XBP1s controls cell differentiation, adaptation, survival, and cell identity of highly secretory cells such as hepatocytes, pancreatic acinar and β-cells, and intestinal goblet cells [36,39,40], and the increased levels of protein synthesis and secretion make these cell types vulnerable to ER stress.

IRE1-mediated degradation of mRNAs, known as regulated IRE1-dependent decay, helps reduce ER stress by decreasing the abundance of mRNAs and synthesized proteins arriving to the ER for protein folding, including degradation of transcripts that may promote apoptosis such as death receptor 5 [41,42]. However, during unmitigated ER stress, the phosphorylated IRE1α appears to favor a switch from homodimers to higher oligomers, increasing the affinity of its RNase domain to additional RNA substrates. This leads to depletion of ER protein folding components, which further exacerbates the ER stress [43]. Under conditions of persistent UPR activation, IRE1α RNase also degrades microRNAs (miRs-17, -24a, -96, and -125b) that normally repress translation of caspase 2 mRNA, promoting caspase 2 activation and cell death [44].

Activation of IRE1 is not restricted to ER stress and the UPR. Through its cytoplasmic domain, IRE1 located at MAMs can be activated by docking signaling competent factors, independently of ER stress. This activation helps in regulating the redistribution of IP_3_Rs and the local transfer of Ca^2+^ from the ER to the mitochondria matrix [11]. Subcellular distribution of IRE1 and EIF2AK3/PERK at MAMs has been suggested to optimize Ca^2+^ signaling and the crosstalk between these organelles [12]. In addition, the physical interaction of IRE1 with TNFα receptor-associated factor (TRAF2) activates NF-κB and c-Jun N-terminal kinase (JNK), thereby inducing inflammatory mediators. Through ER stress, inflammatory stimuli, or engagement of pattern recognition receptors (PRRs), IRE1-XBP1s signaling in myeloid cells controls eicosanoid metabolism, biosynthesis of prostaglandins (e.g., PGE2), and the resultant pain from tissue injury [45]. Finally, IRE1 can physically interact with proapoptotic proteins such as Bcl-2-associated X protein (BAX/BCL2L4) and Bcl-2-antagonist/killer 1 [46]. These interactions may alter the ER-mitochondria Ca^2+^ balance and subsequently induce mitochondrial-dependent cell death [3,46]

### 2.2. PERK

Upon ER stress, the EIF2AK3/PERK phosphorylates the eukaryotic translation initiation factor 2 alpha subunit (eIF2α) at serine 51, inhibiting protein synthesis. This action mitigates the ER stress while maintaining the translation of mRNA molecules that favor the UPR [15,47]. In this manner, the EIF2AK3/PERK-eIF2α pathway upregulates activating transcription factor 4 (ATF4), which appears to have dual functions. On one hand, it increases the biosynthesis of amino acids, chaperones, foldases, and components of the ERAD machinery to enhance ER function, mitigate the ER stress, and maintain cellular homeostasis [47]. ATF4 also upregulates protein phosphatase 1 (PP1) and the growth arrest and DNA damage-inducible protein (GADD34), which dephosphorylate and activate eIF2α, restoring protein synthesis [12]. On the other hand, EIF2AK3/PERK-eIF2α-ATF4 induces the transcription of CHOP (CCAAT/enhancer-binding protein homologous protein), leading to apoptosis during prolonged or unmitigated UPR. Of note, under established ER stress, restoration of mRNA translation by GADD34, or the expression of death receptor 5 by CHOP [42] can aggravate the dysfunctional UPR, leading to apoptosis [10,48,49]. EIF2AK3/PERK is found at MAMs, where it regulates reactive oxygen species (ROS) propagation under ER stress [11], supporting the idea that persistent EIF2AK3/PERK activation and Ca^2+^ release from the ER promotes mitochondrial damage and cell apoptosis [50].

CHOP enhances the expression of the ER oxidase 1α, which induces ROS-mediated oxidative damage and Ca^2+^ release from the ER by activating IP_3_Rs. Ca^2+^ released from the ER-storage proteins into the cytosol reaches the mitochondrial membrane, promoting oxidative damage and resulting in the release of c-cytochrome and the assembly of the apoptosome [4]. Additionally, ROS, apart from activating Ca^2+^ release from the ER, are also considered to act as signaling molecules by regulating the activity of protein kinases and protein phosphatases. For instance, ROS can activate NF-κB and JNK and subsequently promote inflammatory and apoptotic signaling in the UPR [4]. CHOP activates proapoptotic proteins Bim (BCL2L11), telomere repeat binding factor 3, and death receptors, while inhibiting the prosurvival factor, Bcl-2 [51]. The promotion of apoptosis by CHOP aligns with the low level of both apoptosis and inflammation in the colon of dextran sulfate sodium (DSS)-treated CHOP^−/−^ mice [52].

EIF2AK3/PERK also phosphorylates the nuclear factor-erythroid-2-related factor 2 (NRF2) [53], which, together with ATF4, controls the expression of antioxidant proteins (e.g., oxidoreductases, glutathione-S-transferase, and phenolic sulfotransferases). These proteins counteract the effects of ROS and promote cell survival [53,54].

Moreover, eIF2α can also be phosphorylated by eukaryotic translation initiation factor 2 alpha kinase 2 (EIF2AK2), also known as double-stranded RNA-activated protein kinase (PKR) [55]. This kinase is usually activated by viral infection and pathogen-associated molecular patterns (PAMPs), such as lipopolysaccharide (LPS) and inflammatory cytokines (e.g., TNFα). A role of EIF2AK2 in activating the adaptive UPR, prosurvival signaling, and proliferation of intestinal epithelial cells was reported in DSS-colitic EIF2AK2/PKR^−/−^ mice [56]. However, EIF2AK2/PKR-mediated activation of eIF2α was also linked to apoptosis [57,58]. Thus, moderate activation of the EIF2AK3/PERK branch of the UPR can be protective, whereas prolonged activation may promote apoptosis [34].

### 2.3. ATF6α

ATF6α, a type II transmembrane protein within the ER membrane, undergoes proteolytic processing to generate the active bZIP transcription factor ATF6αp50. Under normal conditions, ATF6α stably binds to HSPA5/BiP. However, ER stress stimulates the ATPase activity of HSPA5/BiP, leading to its dissociation from ATF6α [59]. This liberates ATF6α monomers, which then relocate to the Golgi apparatus to be cleaved by site-1 and site-2-proteases in the luminal and transmembrane domains, respectively [37,60,61]. This processing results in the generation of the ATF6αp50 transcription factor.

ATF6αp50 binds to the CCAAT consensus sequence known as the cis-acting ER stress response element in the DNA, initiating the transcription of ER and ERAD-associated genes to expand the ER organelle and its protein folding capacity, including the expression of XBP1, CHOP, and ER chaperones [62]. ATF6αp50 can form heterodimers with XBP1s and synergistically enhance the UPR response [63], favoring the synthesis of proteins essential for folding and degradation, which, in turn, confers cytoprotection.

Both XBP1s and ATF6p50 regulate the transcription of genes encoding ER chaperones (e.g., HSPA5/BiP, HSP90B1/GRP94 and DNAJC3/p58IPK), foldases such as PDI, growth factors including mesencephalic astrocyte-derived neurotrophic factor, and enzymes vital for lipid membrane biogenesis, as well as the modification, translocation, and secretion of proteins [12]. Taken together, ATF6α enacts various effector mechanisms essential for cytoprotection, membrane biogenesis, proper protein folding, and protein secretion to maintain ER-homeostasis.

## 3. ER Stress and UPR in Inflammatory Bowel Disease (IBD)

### 3.1. Homeostasis of the Intestinal Epithelial Barrier

The intestinal mucosa constitutes a barrier with various components, including a thick layer of secreted mucins. The glycocalyx, composed of glycoproteins, glycolipids, and sulfated polysaccharides, resides beneath this mucus layer and above epithelial cells [64]. The barrier further incorporates a single layer of diverse columnar epithelial cells expressing transmembrane mucins and underpinned by a collagen-rich basement membrane. Additionally, leukocytes constantly patrol the epithelium, sampling intestinal luminal content and coordinating immune responses in the lamina propria. The intestinal epithelium integrates signals from the gut microbiota and from resident leukocytes to maintain intestinal homeostasis.

Intestinal epithelial cells (IECs) include at least four types of epithelial cells in both the small and large intestine: intestinal stem cells, tuft cells, absorptive epithelial cells (enterocytes), and a subgroup of secretory cells (including goblet, Paneth, and enteroendocrine cells). Disruption of the intestinal epithelial barrier can precipitate the onset of inflammatory bowel disease (IBD) [65,66]. Such defects in the epithelial barrier may originate from changes in the commensal microbiota composition (particularly dysbiosis favoring protease-producing mucolytic bacteria) [67], dysregulated immune responses, and genetic mutations (e.g., deficiency in α-defensins) [65,66].

IECs are continuously exposed to potential ER stress-inducing factors, including nutrient deprivation (such as glucose and lipid deprivation), alterations to different steps in protein synthesis (folding, glycosylation, and lipidation), and microbial toxins (e.g., LPS) that trigger inflammatory signaling. These factors induce basal levels of ER stress and activate the UPR, which can lead to either adaptation and survival or induction of epithelial cell death via apoptosis [68]. Persistent ER stress in IECs can overwhelm the capacity of the ER to synthesize the proteins required for major cell and organ functions, ultimately triggering a pathological UPR.

Besides nutrient and water absorption, the main functions of colon epithelial cells, indicated by the large number of goblet cells in this tissue, involve mucin production. Maintenance of a functional mucus layer is crucial for three main reasons: (1) forming a physical barrier to prevent commensal bacteria from infiltrating the lining epithelium while permitting the passage of low-molecular-weight soluble luminal antigens and nutritional factors; (2) maintaining lubrication for the colonic mucosa to facilitate the mechanical transport of feces; and (3) providing a first-line innate immunity lattice/matrix where secreted proteins such as immunoglobulins [69,70], antimicrobial peptides (AMP), and enzymes (e.g., lysozyme, defensins, cathelicidins, lipocalins, and C-type lectins) can be anchored to mediate neutralization effects [65]. In sum, certain levels of signals originating from the gut microbiota, the intestinal epithelium, and the innate immunity cells in the lamina propria are sensed as ER stress by IECs and immune cells, resulting in UPR activation to maintains intestinal homeostasis.

### 3.2. Pathogenesis of IBD

IBD, which encompasses ulcerative colitis (UC) and Crohn’s disease (CD), stems from multiple etiologies [65,68]. UC is mainly characterized by mucosal inflammation in the colon and rectum, whereas CD shows transmural inflammation in any region of the small and large intestine [71]. At least three pathogenic mechanisms have been proposed for IBD: (1) induction of IEC apoptosis; (2) disruption of mucosal barrier function; and (3) induction of proinflammatory responses in the gut [38]. In addition, abnormalities in the ER stress response have been proposed as an alternative mechanism for IBD pathogenesis [72,73].

#### 3.2.1. Apoptosis of IECs

Apoptosis is a physiological catabolic process that is crucial for maintaining the homeostasis of the intestinal mucosa [66]. Increased apoptosis of colon epithelial cells has been observed in UC patients [74]. Apoptosis can be activated via various mechanisms, including signaling through death receptors, such as FAS cell surface death receptor and tumor necrosis factor receptor (TNFR) family members, and via mitochondrial damage and caspase activation [66]. In addition, unmitigated ER stress and UPR may lead to apoptosis [38]. Expression of Raf kinase inhibitor protein (RKIP) and p53-upregulated modulator of apoptosis (PUMA) is increased in human colon tissue and correlates positively with the severity of IBD in humans and colitis in mice [75,76]. Colon tissue from UC patients also shows increased levels of caspase recruitment domain (CARD9, CARD14, and CARD15) family members, mediators of apoptosis [77]. Furthermore, increased frequencies of FasL- and perforin-positive cells in the lamina propria have been observed in colon tissue from IBD patients [74]. Finally, apoptosis in IBD may involve mechanisms beyond the activation of canonical apoptosis pathways. For example, high-mobility group box 1 (HMGB1) and the human antigen R (HuR, ELAVL1) proteins have been found to mediate apoptosis in UC [78].

#### 3.2.2. Disruption of the Mucosal Barrier

ER stress in IECs appears as a result of overwhelming stimulation to maintain barrier function (for instance, AMP production by Paneth cells and mucin production by goblet cells); this may lead to cell death and disruption of the mucosal barrier [38]. Mucins, which are large and highly O-glycosylated proteins that are produced abundantly, require complex posttranslational modifications, including glycosylation, disulfide bond formation, and oligomerization. For this reason, they have an increased likelihood of being misfolded during biosynthesis [79]. IRE1β regulates the rate of mucin biosynthesis, folding, and secretion by controlling *MUC2* mRNA transcript levels. IRE1β deficiency causes ER stress and increases susceptibility to colitis [80], as does the misfolding of mucins in *Muc2*^−/−^ mice [81].

#### 3.2.3. Excessive Inflammation

IBD may arises from excessive inflammation, marked by the overproduction of inflammatory mediators and increased apoptosis of epithelial cells [66,75,77,82]. UC patients show increased expression and constitutive activation of toll-like receptor 4 and nuclear factor kappa B (TLR4/NF-κB) in colon epithelial cells that may progress into colitis associated cancer [83,84,85]. ER stress in IECs amplifies their response to TLR5 ligand Flagellin by producing IL-8 cytokine that impacts the activation of dendritic cells and promotes inflammation [86]. Moreover, activated leukocytes in the lamina propria of UC patients release proinflammatory cytokines and increase the production of ROS, exacerbating the apoptosis of epithelial, immune and bystander cells [66]. Under these conditions, the anti-inflammatory effects of IL-10 are attenuated, which typically upregulate genes required for ER stress mitigation, ERAD-mediated misfolded protein degradation, and enhanced MUC2 folding and secretion [87].

Sentinel goblet cells located at the entrance of colonic crypts express TLRs and Nucleotide binding oligomerization domain (NOD)-like receptors capable of sensing PAMPs. These cells activate the NLRP6 inflammasome to secrete mucins [88]. This response in the colon is similar to that in Paneth cells in the small intestine, which respond to TLR ligands by secreting AMP and defensins [89]. In the absence of those receptors or the MyD88 adaptor protein, mice display increased susceptibility to bacterial colonization and DSS colitis [65,90]. This supports the notion that tonic microbial signals induce the secretion of mucins and defensins to maintain intestinal homeostasis. However, persistent stimulation of PRRs can lead to chronic inflammation [65].

#### 3.2.4. Aberrant UPR Activation

ER stress in IECs, including goblet cells, may also stem from genetic defects leading to aberrant UPR activation. The ileum and colon tissues of CD and UC patients show elevated levels of UPR activation compared to the normal tissues of healthy individuals, associated with increased expression of a chaperone, HSPA5/BiP, and a transcription factor, XBP1s [72]. Genetic defects in mice—for instance, those deficient in IRE1β or those with *MUC2* mutations (*Winnie* and *Eeyore* mice) caused by carcinogen (N-ethyl-N-nitrosourea) exposure—result in aberrant MUC2 biosynthesis. This, in turn, induces ER stress and UPR, developing a UC-like phenotype [80]. Goblet cells in these mice accumulate misfolded MUC2 protein and its precursor, triggering ER stress, which leads to reduced MUC2 secretion and changes in the viscoelastic properties and composition of the mucus barrier. This favors inflammation, with enhanced local production of inflammatory cytokines (e.g., IL-1β, TNFα, and IFNγ) [81]. These proinflammatory cytokines also induce ER stress in intestinal epithelial cells to augment the expression and secretion of mucins [70], which not only exacerbates the ER stress-mediated mucin depletion but also leads to the disassembly of tight junction and adherence junction proteins, accelerating the epithelial layer leakiness [91,92]. Together, the epithelial barrier defects and inflammation progress to dysregulated UPR and persistent inflammation, both common features of IBD [79,81].

UC patients show a progressive loss/dysfunction of goblet cells [93]. MUC2, produced by goblet cells, plays a prominent role in colon homeostasis by maintaining appropriate levels of cell proliferation and apoptosis, regulating cell migration along the crypt, and suppressing inflammation and colorectal cancer [94]. Accordingly, *Muc2*^−/−^ mice develop aberrant crypt morphology and exhibit altered maturation and migration of epithelial cells, which eventually transform into adenomas and adenocarcinomas due to an increased rate of cell proliferation over apoptosis [94]. A similar phenotype is observed in mice lacking the protein complexes required for post-translational processing of MUC2, such as glycosylation [95], sulfonation [96], disulfide bond formation [97], and lipidation/palmitoylation [98]. Consequently, proper MUC2 folding and secretion are required for colon crypt homeostasis. An increased demand in mucin secretion may cause misfolded MUC2, which, in turn, induces ER stress and UPR, contributing to the pathogenesis of UC.

Nonetheless, the pathogenesis of IBD is not completely understood, and there may be overlooked factors involved in intestinal homeostasis, whose presence or disruption could cause ER stress, thereby exacerbating the manifestation of IBD. For example, nutrient deprivation (glucose, lipids, or amino acids) induces ER stress, which subsequently activates autophagy. This mechanism generates increased levels of ROS that stimulate mucus secretion by intestinal goblet cells and contribute to the elimination of severely damaged cells to maintain colon homeostasis [99,100]. However, excessive production of ROS may cause cell death.

## 4. Pharmacologic Intervention of the UPR as a Potential Therapeutic Strategy in IBD Treatment

Given the protective roles of the intestinal epithelium often dysregulated in IBDs, such as secretion of the mucin-rich mucous layer and AMPs, interaction and sensing of the normal gut microbiota, and transmission or amplification of those signals to intraepithelial lymphocytes and lamina propria resident leukocytes, it would be beneficial to develop novel therapeutics targeting the restitution and functionality of epithelial cells for IBD treatment. Thus, new therapies could focus on (1) enhancing colon crypt goblet cell function (e.g., mucin folding and secretion); (2) promoting survival and proliferation of colon crypt stem cells and other proliferation-competent crypt cells such as transit-amplifying cells; and (3) integrating both strategies, namely, the survival/proliferation of colon crypt stem cells and their differentiation into goblet and other terminally differentiated epithelial cells.

The ER stress and activation of the UPR in eukaryotic cells is a mechanism that initially promotes cell homeostasis, even though chronic activation can lead to cell death. In the context of IBD, this notion is supported by findings that IRE1β^−/−^ mice exhibited increased susceptibility to experimental colitis, indicative of a protective role of IRE1 of the UPR in the intestinal epithelium [72,73]. Similarly, *Xbp1* knockout mice displayed increased ER stress in small intestine and colon epithelial cells, associated with the spontaneous inflammation and dysfunctional secretion of AMP by Paneth cells and mucin by goblet cells [72]. The heightened colon inflammation and susceptibility to DSS-induced colitis in mice deficient in ATF6α or chaperone genes highlight the role of this UPR transcription factor and chaperones during colitis [101]. ATF6α expression is induced by IFN-γ, with IFN-γ-ATF6/CEBPB signaling being necessary to initiate autophagy to combat bacterial infections, a situation exacerbated in ATF6α^−/−^ mice [102].

Consequently, UPR components are promising targets for therapeutic intervention in ER stress- and UPR-associated pathologies. Several inhibitors targeting the IRE1 kinase activity or dephosphorylation of eIF2α have been developed and tested in in vitro studies of ER stress and UPR modulation [103,104,105]. Table 1 summarizes the strategies targeting the UPR for IBD treatment. Targeting the UPR elements, including the KDELRs, has been proposed either to activate and induce ER stress response or to inhibit and suppress ER stress sensors that favor cell survival/proliferation (organ preservation) [15,23,33]. Alternatively, a strategy that overwhelms the UPR by using proteasome inhibitors has been employed to induce apoptosis of malignant cells [106,107]. A range of natural and synthetic molecules, some of which impact the UPR, have been explored in IBD treatment and reviewed recently [108]. However, limited research has been conducted on recombinant pharmaceutical proteins designed to modulate the UPR towards the induction of cell survival/proliferation to restore cell homeostasis. Hereafter, we summarize those compounds found to modulate the UPR in IBD.

### 4.1. Targeting the UPR Sensors

A synthetic XBP1 agonist, HLJ2, was generated based on a monomeric compound purified from species of *Coptis* and *Corydalis* herbaceous plants. HLJ2 reduced inflammatory cytokines (e.g., TNFα, IL-1β, and IL-6) in DSS colitis mice. Additionally, HLJ2 increased the expression of tight junction proteins ZO-1 and claudin-1 in colon tissue, alleviated intestinal dysbiosis, and protected the intestinal mucosa [114]. Thus, HLJ2 is postulated to reinforce the colon epithelial barrier and reduce inflammation.

Salubrinal, a small molecule identified to block the phosphatases mediating eIF2α dephosphorylation, was found to enhance cell survival/cytoprotection in cells under ER stress induced by tunicamycin or viral infection, by maintaining levels of eIF2α and protein synthesis [105]. In DSS-induced colitis, salubrinal showed protective effects by suppressing the expression of proinflammatory cytokines and myeloperoxidase activity, while elevating ATF4 and HSPA5/BiP, which ameliorated ER stress and improved histological scores of colitis [110].

### 4.2. Non-Specific Targeting of the UPR

The ER stress and the UPR in colon epithelial cells and colitis have been non-specifically targeted with single amino acids (e.g., glutamine) [109], hormones (e.g., estrogen) [113], and chemical chaperones such as tauroursodeoxycholic acid (TUDCA) and 4-phenylburyrate (PBA) [101,108]. The PBA and TUDCA chaperones reduce ER stress in IECs and mitigate features of acute and chronic colitis induced by DSS or generated by deficiency in the anti-inflammatory IL-10 cytokine (IL10^−/−^ mice) [101]. The estrogen receptor agonist, G-1 (1-[4-6-bromobenzol [1,3] dioxol-5yl]-3a,4,5,9b-tetrahydro-3H-cyclopenta[c]quinoloin-8-yl]-ethanone), acting through the G-protein-coupled estrogen receptor (GPER) at the plasma membrane, reduced the expression of HSPA5/BiP and CHOP, and attenuated all three arms of the UPR in DSS-colitis mice. In addition, G-1 reduced the level of cleaved caspase-3 and increased the number of Ki-67+ proliferating colon crypt epithelial cells [113].

Artesunate, a hemisuccinate derivative of artemisinin, isolated from *Artemisia annua* herbs, has demonstrated protective effects in colitis [111]. Artesunate inhibited ER stress signaling pathways involving PERK-eIF2α-ATF4-CHOP and IRE1α-XBP1 during DSS-induced colitis. Artesunate inhibited NF-κB and proinflammatory cytokines, improved clinical and histopathological scores, and maintained the levels of claudin-1 and MUC2 in the colonic mucosa [112].

Finally, ghrelin, a nutrient sensor expressed in epithelial and immune cells, acts through the growth hormone secretagogue receptor (GHS-R1a) and represents another potential UPR modulator in the gastrointestinal tract. Ghrelin was found to modulate the UPR by reducing the expression of proapoptotic factors including CHOP, caspase 3, and BAX while increasing Bcl-2, which provides survival signals. Additionally, ghrelin inhibited TNFα-induced apoptosis in Caco2 cells and protected colon epithelial cells from DSS and TNBS-induced colitis [82].

### 4.3. EPICERTIN, a KDELR Ligand Modulator of the UPR

Recombinant KDEL-tagged fusion proteins have been engineered for therapeutic intervention in cancer and for vaccine development against virus-induced tumors [117,118] and also as potential subunit vaccines against *Vibrio cholerae* [119]. EPICERTIN consists of the nontoxic, recombinant cholera toxin B-subunit (CTB) modified with a KDEL motif in its C-terminal region, exhibiting wound healing properties in the colon [115]. The potential of EPICERTIN as a therapeutic agent for IBD has been investigated in mouse models of colitis, where the KDEL motif has been found to be crucial for its wound healing activity [116]. EPICERTIN bears two structural modifications distinguishing it from CTB: (a) Asn4 → Ser mutation to avoid *N*-glycosylation when expressed in eukaryotic cells; and (b) a C-terminal hexapeptide containing the KDEL ER retrieval sequence [119]. EPICERTIN retains GM1-binding affinity, molecular stability, and vaccine efficacy, as demonstrated by the induction of anti-toxin IgG and IgA antibodies upon oral immunization in mice. The conformational structure model of EPT closely resembles that of CTB [120] (Figure 1A).

EPICERTIN binds to GM1-ganglioside at the plasma membrane of colon epithelial cells, undergoes endocytosis, and is retrogradely transported to trans Golgi network and the ER via a lipid-based sorting pathway [121,122] or by interacting with KDELRs at the plasma membrane [123]. Once in the ER, EPICERTIN displaces sensor-bound chaperones to induce the UPR (Figure 1B). EPICERTIN, unlike CTB or the C-terminal leucine-truncated EPICERTIN variant (“CTB-KDE”), possesses unique mucosal wound healing activity by increasing TGFβ1 and TGFβ2 levels, a cytokine and growth factor known to induce IEC migration and restitution [124] and found to facilitates Caco2 cell migration and wound healing in scratch assays [116]. In in vitro cultures of colon tissue explants from IBD patients, EPICERTIN promoted colon crypt survival and upregulated wound healing pathway genes, including *TGFB1*, *CDH1*, and *WNT5A* [116]. The wound healing effects of EPICERTIN were corroborated in in vivo mouse models of acute and chronic colitis induced by single and multiple DSS exposure cycles [125], respectively. EPICERTIN decreased histopathological scores and reduced inflammatory cytokine levels, while increasing innate immune cell populations, including dendritic cells, natural killer cells, and macrophages (both M1 and M2) in the colon [115]. In addition, EPICERTIN, orally administered once every two weeks (four doses), significantly reduced tumor development in an azoxymethane/DSS model of colitis-associated cancer.

In mice, EPICERTIN remained detectable in the colon epithelium 24 h after oral or intrarectal administration, particularly in the base region of colon crypts. This suggests a role for colonic crypt base stem cells in EPICERTIN-induced epithelial regeneration [126]. Mechanistically, the IRE1/XBP1s pathway of the UPR, which is known to maintain ER homeostasis by regulating cell survival/apoptosis balance [33,127], appears to mediate the wound healing activity of EPICERTIN, as the IRE1 inhibitor 4µ8C or siRNA knockdown of *Xbp1* abolished EPICERTIN’s wound healing effect in Caco-2 cells [116]. Collectively, these findings suggest that EPICERTIN activates the IRE1/XBP1 pathway, which, in turn, appears to promote epithelial cell restitution via secretion of TGFβ and other wound healing-related growth factors. Although the link between IRE1 activation and mucosal epithelial repair is not fully understood, these studies have demonstrated the potential therapeutic application of EPICERTIN, a KDELR ligand and an UPR-modulating molecule, in colon epithelial cells. It remains to be investigated whether TGFβ is the sole signaling mechanism mediating the healing effects of EPICERTIN in both acute and chronic colon inflammation. In addition, the effects of EPICERTIN on leukocytes in the colon lamina propria [115] and their potential contribution to mucosal healing remain to be disclosed.

**Figure 1 biomedicines-11-02066-f001:**
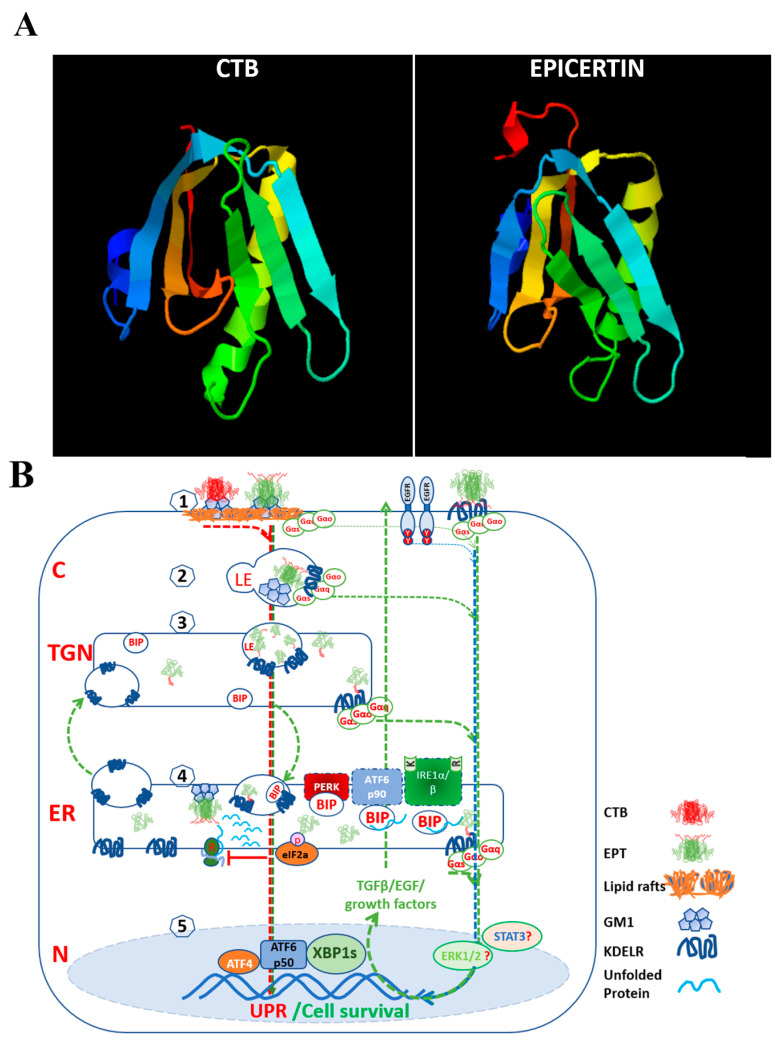
EPICERTIN: A KDELR ligand and a modulator of the unfolded protein response that promotes cell survival and proliferation. (**A**) Structural models of CTB and EPICERTIN (EPT) monomers generated by the Protein Homology/analogy Recognition Engine V 2.0 (Phyre^2^) software created by the Structural Bioinformatics Group, Imperial college, London and updated on January 25, 2021 [120] based on a crystal structure of CTB (Protein Data Bank ID: 3CHB). The C-terminal KDEL amino acid sequence extension of EPT is highlighted in red at the top of the image. (**B**) A proposed model of EPICERTIN-KDELR signaling and modulation of the UPR. 1. Both CTB and EPT bind to GM1 at the plasma membrane, but with differing outcomes. EPT may also bind to KDELR on the plasma membrane. 2. CTB and EPT are endocytosed in early endosomes, and eventually, the pentamer disintegrates into monomers within late endosomes. 3. EPT and CTB are retrogradely transported to the trans-Golgi network (TGN) and finally to the ER, following a lipid-based sorting pathway. Only EPT interacts with and activates KDELR signaling, promoting vesicular transport and the recycling of ER chaperones. Activation of KDELR by EPT may also occur early on the endosome membrane and in the TGN. 4. In the ER, both CTB and EPT may displace chaperones from ER stress sensors (PERK, ATF6α, IRE1α/β) and induce the unfolded protein response (UPR), although only EPT is known to activate IRE-XBP1s. 5. Activation of the ER stress sensors leads to the generation of the transcription factors phosphorylated eIF2α, ATF6αp50, and XBP1s, which transactivate genes in the nucleus that are associated with ER homeostasis. The increased endocytosis of CTB may trigger apoptosis by various mechanisms, including a chronic UPR or activation of death receptors [128]. In contrast, EPT–KDELR–chaperone complexes are allowed to recycle between ER and TGN, alleviating the ER stress by replenishing chaperones back into the ER, modulating the intensity and amplitude of UPR. Alternatively, the surplus of endocytosed CTB eventually overwhelms the endosome-to-TGN trafficking, which is known to activate the NLRP3 inflammasome to induce apoptosis [129]. EPT–KDELR interaction might activate signaling pathways regulating and improving vesicular trafficking and cell survival which are currently unknown; however, preliminary data suggest the involvement of MAPK kinases (unpublished observations). C: Cytoplasm; TGN: Tans-Golgi Network; ER: Endoplasmic Reticulum; N: Nuclei; L: Lysosome; LE: Late Endolysosome.

## 5. Concluding Remarks

Our growing understanding of ER stress and the specific regulation of UPR that occurs in different types of eukaryotic cells is advancing towards a point where therapeutic manipulation of the UPR may be viable to address human pathologies associated with ER perturbation, such as IBD. Although results derived from various IBD models that demonstrate therapeutic effects through UPR manipulation are currently limited, there is potential for properly reversing ER-related pathological conditions associated with the disease. EPICERTIN emerges in this context as a promising UPR modulator that promotes cell survival, migration, and the proliferation of epithelial cells, ultimately leading to epithelial restitution, a therapeutic aim not yet addressed by current IBD treatments. EPICERTIN induces an adaptive UPR and stimulates the production and release of TGF-β, which, in turn, promotes epithelialization. Further investigation into EPICERTIN’s mode of action will illuminate the protective role of UPR in epithelial wound repair and pave the way for a potential new therapy for IBD.

## Figures and Tables

**Table 1 biomedicines-11-02066-t001:** Therapeutic strategies in inflammatory bowel disease targeting the unfolded protein response.

In Vivo/In Vitro Model	ER Stress Inducer	Targeted UPR Sensor	Drug Candidate	Mode of Action	References
Mouse model of DSS-induced colitis. P58 (IPK^−/−^), ATF6a^−/−^, IL10^−/−^ mice, and IEC-6 cells	DSS-induced colitis/cytokine cocktail (TNF-α, MCP-1, IL-1β)	Induction of ATF6α- and chaperone P58; reduce ER stress	Phenyl butyric acid (PBA), Taurine-conjugated ursodeoxycholic acid (TUDCA)	Promote protein folding; reduce features of acute and chronic colitis; reduce BiP, peIF2a, CHOP, and cleaved caspase 3/12.	Cao et al., 2013 [101]
Rat model of TNBS-induced colitis; Caco2 cells	TNBS-induced colitis. Brefeldin A and Tunicamyin-induced ER stress	PERK-CHOP, ATF4, ATF6, XBP1s	Glutamine	Reduce CHOP, BiP, Caspases 12, 9, 8, 3. calpain-1, and pJNK.	Crespo et al., 2012 [109]
Mouse model of DSS-induced colitis	DSS-induced colitis	Phosphatase inhibitor Increase eIF2α phosphorylation	Salubrinal	Increase BiP, ATF4, and HSP70; reduce CHOP; suppress MPO, TNF-α, and IL-1β.	Okazaki et al., 2014 [110]
Mouse model of DSS-induced colitis	DSS-induced colitis. ER stress	Prevents activation of PERK-eIF2α-ATF4-CHOP and IRE1α-XBP1 signaling pathways	Artesunate	Increase Bcl-2/Bax ratio; suppress NF-κB p65 and IκBα, IL-1β, IL-6, and TNF-α; increase IL-10; inhibit expression of cleaved-caspase 3.	Yin et al., 2020, 2021 [111,112]
Mouse model of DSS-induced colitis; CCD841 cell line	DSS induced colon ER stress/Thapsigargin	Inhibited PERK, IRE1α, ATF6, reduced BiP, CHOP	G-1	Inhibit apoptosis, increased Ki-67+, and BrdU+ crypt cells; reverse the decrease in cyclin D1 and B1.	Wang et al., 2021 [113]
Mouse model of DSS colitis model	DSS-induced colitis. ER stress	XBP1 agonist, Coptisine-derivative	HLJ2	Decrease MPO, TNFα, IL-1β, and IL-6; increase the expression of ZO-1 and claudin-1.	Zhang et al., 2017 [114]
Mouse model of DSS-induced acute and chronic colitis. Caco2 cells scratch assay	DSS-induced ER stress	Induce IRE1-XBP1s axis.	EPICERTIN	Binds to KDELRs and may promote vesicle trafficking; promotes mucosal wound healing mediated by TGFβ, CDH1, and WNT5.	Baldauf et al., 2017 [115]; Royal et al., 2019 [116]

## Data Availability

Not applicable.

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
