# Peer review of "The Unfolded Protein Response and Its Implications for Novel Therapeutic Strategies in Inflammatory Bowel Disease"

_biomedicines, 2023, doi:10.3390/biomedicines11072066_

Round 1

Reviewer 1 Report

Noel Verjan Garcia et al is an interesting review article discussing the challenges in the treatment for IBD which does not exist to date. The author’s recent studies have shown that EPICERTIN, a recombinant variant of the cholera toxin B subunit containing an endoplasmic reticulum (ER) retention motif, can induce a protective unfolded protein response (UPR) in colon epithelial cells, subsequently promoting epithelial restitution and mucosal healing in IBD models. These findings support the idea that compounds modulating UPR may be promising pharmaceutical candidates for not only managing the symptoms but treatment of the IBD. In this review, the authors have summarized the current understanding of ER stress and UPR in IBD, focusing on their roles in maintaining cell homeostasis, dysregulation, and disease pathogenesis. They also discuss therapeutic strategies that promote cytoprotection of colon epithelial cells to reduce inflammation via pharmacological manipulation of the UPR. In addition, abnormalities in the ER stress response have been proposed as an alternative mechanism for IBD pathogenesis.

The discussion herewith presented is impactful and relevant regarding pharmaceuticals and IBD management and treatment attempts. Given the protective roles of the intestinal epithelium often dysregulated in IBDs, such as secretion of the mucin-rich mucous layer and AMPs, interaction and sense of the normal gut microbiota, and transmission or amplification of those signals to intraepithelial lymphocytes and lamina propria resident leukocytes, it would be beneficial to develop novel therapeutics targeting the restitution and functionality of epithelial cells for IBD treatment. Thus, new therapies could focus on i) enhancing colon crypt goblet cell function, ii) promoting survival and proliferation of colon crypt stem cells and other proliferation-competent crypt cells such as transit-amplifying cells, and iii) integrating both strategies, namely the survival/proliferation of colon crypt stem cells and their differentiation into goblet and other terminally differentiated epithelial cells.

There is a growing understanding of ER stress and the specific regulation of UPR that occurs in different types of eukaryotic cells is advancing towards a point where pharmaceutical therapeutic manipulation of the UPR may be viable to address human pathologies associated with ER perturbation, such as IBD. Unfortunately, the results derived from various IBD models that demonstrate therapeutic effects through UPR manipulation are still limited, there is potential for properly reversing ER-related pathological conditions associated with the disease. Further, the results obtained from the DDS-induced colitis in mice model (Table 1) may not represent the same in human-IBD colitis. DDS colitis is not inflammatory in nature, rather, is trauma/injury caused by chemical toxicity. In humans IBD, inflammation is a result of an antibody-antigen reaction against the mucosal resistance of an individual patient.

In general, I enjoyed reading this well-summarized review. Good job.

Reviewer 2 Report

The article is very interesting because it concerns the pathomechanism of inflammatory bowel diseases. It is worth noting that Sentinel goblet cells located at the entrance of colonic crypts express TLRs and Nucleotide binding oligomerization domain (NOD) like receptors capable of sensing PAMPs. These cells activate the NLRP6 inflammasome to secrete mucins. This response in the colon is similar to that in Paneth cells in the small intestine, which respond to TLR ligands by secreting AMP and defensins 

secreting AMP and defensins. Furthermore, given the protective roles of the intestinal epithelium, often dysregulated in IBD, such as the secretion of mucin- and AMP-rich mucosa, the interaction and sensing of the normal intestinal microbiota, and the transmission or amplification of these signals to epithelial cells and lamina propria resident leukocytes, it would be beneficial to remove development of new drugs targeting the restitution and functionality of epithelial cells for the treatment of IBD. Thus, new therapies may focus on: 

enhancing colonic crypt goblet cell function (e.g., mucin folding and secretion), 

 promoting survival and proliferation of colonic crypt stem cells and other proliferating crypt cells, such as transit ampli cell splicing, and 

 integrating both strategies, namely colon crypt survival/proliferation and their differentiation into cups and other terminally differentiated epithelial cells.